# Distribution and Evolution Law of Void Fraction in the Goaf of Longwall Mining in a Coal Mine: Calculation Method and Numerical Simulation Verification

**Jiaqi Wang [1], Nan Zhou [1,\*], Chongjing Wang [2], Meng Li [3] and Guohao Meng [1]**

[1] School of Mines, China University of Mining and Technology, Xuzhou 221116, China; 13852037975@163.com (J.W.); mengguohao@cumt.edu.cn (G.M.)
[2] Jining Energy Development Group Co., Ltd., Jining 272073, China; CUMTcjwang@126.com
[3] State Key Laboratory of Coal Resources and Safe Mining, China University of Mining and Technology, Xuzhou 221116, China; limeng1989@cumt.edu.cn
[\*] Correspondence: 5724@cumt.edu.cn

**Abstract:** Many voids are produced in the mining process of ore-bearing strata. To explore the development law of voids after mining coal-bearing strata, a theoretical model was established to derive the overall distribution and shape of voids in the goaf. The above theory was verified using the numerical calculation method. The turning point of the void change was found. The research results show that the void in the goaf was widely distributed around the stope, and the overall void ratio was affected by the mining conditions, such as the mining height and face length. While advancing the working face, the dynamic development of the void first increased and then decreased. At first, the distribution of the void ratio in the goaf was between 0.293 and 0.889 under specific geological conditions, and then, with the advancement of the working face, a large void ratio was reserved at 0~40 m behind the working face. When the working face was advanced to the first roof collapse length, the void fractures continued to decline. Using the above voids, the backfilling of solid mine waste can be effectively realized, and the ecological environment can be protected.

**Keywords:** goaf; voids; numerical simulation; gangue; waste management

## 1. Introduction

Coal seam mining destroys the overlying strata's original stress balance. The overlying strata produce tensile damage, shear damage, bending, collapse, and other damage and deformations, thus forming goaf voids and separation gaps in the strata [1–3]. These voids are widely distributed in collapse zones and fracture zones. Due to residual voids in the stope, water and gas gather, leading to water and gas outburst accidents [4–6]. In addition, these voids also lead to the secondary settlement of the goaf, which leads to the settlement of ground buildings, local cracks, tilt, and other damages [7–9]. However, using space as an underground space resource is an environmental protection idea [10–12].

With coal mining, a large amount of solid waste, i.e., gangue, is also produced. The emissions from gangue account for approximately 10~15% of those from coal resources. A large amount of waste rock is directly discharged and piled up on the surface to form a waste rock hill. While polluting and occupying land, air, and water resources, coal mining also seriously threatens the mining area's ecological security and the coal industry's sustainable development. According to China's policy requirements for the construction of safe and efficient green coal mines, the safe mining of deep coal resources and the large-scale treatment of solid coal gangue waste have become important tasks for developing coal enterprises.

Many scholars have analyzed the structure and shape of the overburden in the goaf, the distribution of residual voids in the goaf, and the related residual settlement and stability.

Some scholars [13–15] believe that the overburden structure of the goaf is mainly divided into a caving zone, a fracture zone, and a bending subsidence zone. The rock structure in the caving zone is loose, and the mining structural plane mainly exists in the form of voids, cracks, and some large cavities. Some scholars [16–18] and other researchers have adopted similar material simulation and numerical simulation methods to simulate the existence of "O"-type separation fracture zones around the goaf and put forward the distribution law of fractured rock. They believe that the fracture expansion coefficient of a fractured rock mass is inversely proportional to its distance from the coal seam. Scholars [19–21] have considered the influence of time and goaf span factors, and they have carried out residual settlement predictions for old goaf.

At the same time, some scholars [22–24] have also analyzed the void ratios of stopes using field data. Ma [25] and other scholars have analyzed the void seepage characteristics within the fracture range of a goaf based on the actual measurements, and this provided the equivalent area of the separation fractures after the initial fracture of the key layer. They also analyzed the void permeability coefficients of different fracture development areas. Wang [26] studied a drilling double-plug plugging subsection water injection test device and a drilling television observation system, which were used to detect the overlying rock caving in a fully mechanized mining face, and a digital analysis of the fracture dip distribution characteristics was carried out [27–29]. The authors also used a CMS three-dimensional laser scanning instrument to monitor goaf, and they detected that the goaf within 10 m of the working face had a void with a diameter of 1 to 10 m, as shown in Figure 1. However, only some of the above results could intuitively obtain the distribution forms of goaf voids.

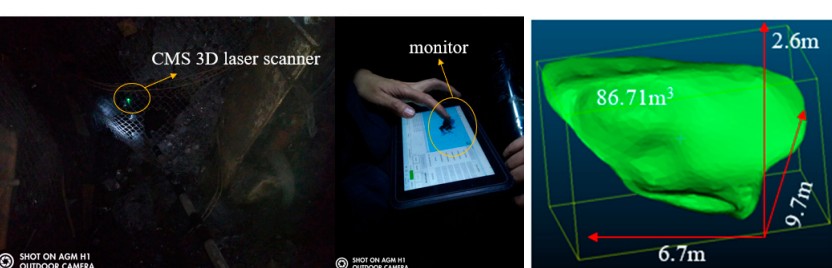

**Figure 1.** Measurement of voids in goaf.

Based on the current research results, this article defines a void fraction as the ratio of a void's volume in goaf to its mining volume. The authors have put forward a concept of the overall distribution of a goaf void, deduced the theoretical formula, and calculated the distribution of the stope gap boundary and the central void ratio. A numerical model was established to study the void development shape of the formed goaf and the dynamic development of the void fraction during the process of advancing. This paper puts forward the utilization method of the void 'gangue fluidization filling method,' expounds its overall concept and system composition, and provides the theoretical basis and ideas for the utilization of goaf gaps and the environmental control of mining areas in China.

## 2. Research Methods

### 2.1. Calculation Method for the Void Fraction

The voids produced by coal mining come from the mined volume. The voids in goaf should be less than the extracted volume of the coal seam, and the bulking volume of the broken direct roof and the bending deformation of the overlying rock prevent the formation of voids. Therefore, the calculation principle of the method was proposed, which was that the volume of voids is equal to the volume of coal seam extraction less the volume of direct roof bulking less the volume of overlying deformation. The calculation model of a void ratio in goaf was established, as shown in Figure 2. In the figure, the upper part of the top plate is composed of a thin plate model, and the lower part is composed of a broken expansion model. The total void ratio of the goaf can be obtained by combining the upper

and lower models. The voids reduced by the deformation of the overburdened rock in the goaf can be calculated by establishing a thin plate model. The voids formed by the broken roof in the goaf can be calculated by establishing a broken roof expansion model.

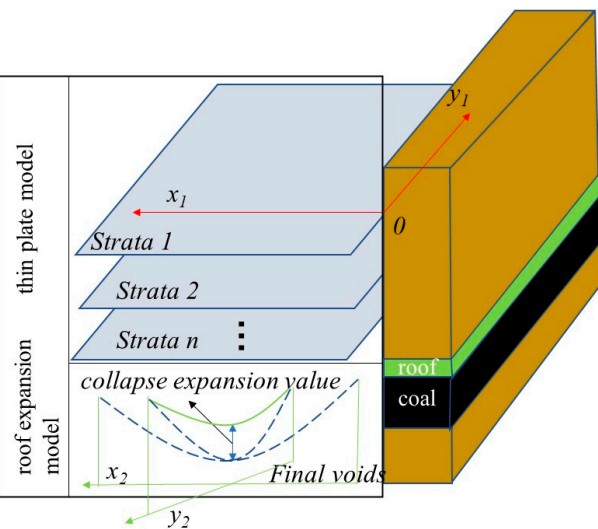

**Figure 2.** Calculation model of the void ratio in goaf.

(1)    Volume of the overlying deformation

Based on the O-ring theory [30–32], the rock strata above a roof are regarded as thin elliptical boundary-fixed plates under the action of a uniformly distributed load. The settlement amount ($w_{ki}$) of any point ($x$, $y$) on a thin plate can be calculated using Formula (1). The coordinate system of the model is shown in Figure 2, and the calculation to determine settlement amount $w_{ki}$ ($x$, $y$) of any point ($x$, $y$) of any rock strata overlying a stope is shown in Formula (2). In the process of the deformation of the overlying strata on a roof, the void ratio of the two adjacent strata can be obtained, according to the different deformations in each layer, as $\varphi_{i, i+1}$ (see Formula (3)).

$$w_{ki}(x,y) = \frac{(\rho_i g T_i + q_{0i}) \cos \alpha \left( \frac{4x^2}{l_x^2} + \frac{4y^2}{l_y^2} - 1 \right)^2}{8 D_i \left( \frac{48}{l_x^4} + \frac{32}{l_x^2 l_y^2} + \frac{48}{l_y^4} \right)} \tag{1}$$

where $\rho_i$ is the rock density of the *i*-th layer; $T_i$ is the thickness of the *i*-th layer; $q_{0i}$ is the upper load of the *i*-th stratum; $\alpha$ is the dip angle of the coal seam; $l_x$ is the strike length of the goaf; $l_y$ is the dip width of the goaf; $D_i$ is the bending stiffness of the *i*-th stratum; and $D_i = ETi^3/(12(1-v^2))$, where $E$ is the elastic modulus and $v$ is Poisson's ratio.

$$w_{ki}(x,y) = \frac{w_{0i}\left(1 - e^{-\frac{x}{2l_i}}\right)\left(1 - e^{-\frac{l_y/2 - |y|}{2l_i}}\right)}{1 - e^{-\frac{l_y}{4l_i}}} \tag{2}$$

and

$$\varphi_{i,i+1} = \frac{\Delta w_{ki} dx dy}{\Delta \sum h_i dx dy} = \frac{w_{ki} - w_{ki+1}}{\sum h_i - \sum h_{i+1}}, \tag{3}$$

where $w_{0i}$ is the subsidence of the *i*-th stratum after it moves and stabilizes; $l_i$ is the rock-breaking length of the *i*-th stratum; $l_y$ is the length of the working face; and $h_i$ and $h_{i+1}$ are the thicknesses of layer $i$ and layer $i + 1$, respectively.

(2)    Volume of voids after direct roof bulking

The void ratio of a broken rock mass below a roof can be expressed by the broken expansion coefficient, as shown in Formula (4):

$$\varphi = 1 - \frac{1}{K_p},\tag{4}$$

where $Kp$ is the coefficient of the crushing expansion of the broken rock and $\varphi$ is the void ratio of the broken rock.

In the $x$-direction, the coefficient of the void's ratio changes in the strike direction of the working face, which can be obtained as follows:

$$K_p = \frac{h_d + H - w_b|(y = 0)}{h_d},\tag{5}$$

where $w_b \,|\, (y = 0) = (H\text{-}h_d(K_{pb}-1))(1\text{-}e^{\text{-}x/2l})$.

By substituting Formula (4) into Formula (5), the void ratio of the collapse zone along the centerline of the floor can be obtained as follows:

$$\varphi_G|(y = 0) = 1 - \frac{h_d}{h_d + H - w_b|(y = 0)},\tag{6}$$

where $\varphi_G \,|\, (y = 0)$ is the void's ratio curve on the central axis of the goaf floor, $h_d$ is the direct top thickness, $H$ is mining height, $w_b \,|\, (y = 0)$ is the subsidence of the basic roof along the central axis of the strike of the floor, $K_{pb}$ is the residual broken expansion coefficient of the fractured rock mass, and $l$ is the length of the broken rock block on the basic roof.

Similarly, in the $y$-direction, the coefficient of the void's ratio changes in the inclined direction of the working face, which can be obtained as follows:

$$\varphi_{G,y} = 1 + e^{-0.15(\frac{l_y}{2} - |y|)}.\tag{7}$$

The relevant literature (Zhang, 2023) has shown that the void ratio change equation for a fractured rock mass under an axial force is

$$\varphi_y = \beta_1\sigma + \beta_0,\tag{8}$$

where $\varphi_y$ is the void ratio of the loose and broken rock under axial stress, $\sigma$ is the relative axial stress (MPa), $\beta_1$ is the regression coefficient, and $\beta_0$ is the void ratio of the broken rock before being subjected to axial stress. When the rock is shale, $\beta_1 = -0.0488$. When the rock is mudstone, $\beta_1 = -0.028$. When the rock is sandstone, $\beta_1 = -0.0254$.

The axial stress ($\sigma$) of the gangue in goaf is determined as follows:

$$\sigma = \frac{(1 - \varphi_G)\gamma(\frac{l_y}{2} - |y|)\sin\alpha}{\sigma_0},\tag{9}$$

where $\sigma$ is the relative compressive stress on any section (MPa), $\gamma$ is the unit weight of the falling rock (N/m³), and $\sigma_0$ is equal to 1 MPa.

Then, the void ratio of the collapse zone is calculated as follows:

$$\varphi_G(x,y) = 1 + \frac{\left[1 + e^{-0.15(\frac{l_y}{2} - |y|)}\right] \cdot \left[1 - \frac{h_d}{h_d + H - [H - h_d(K_{Pb}-1)](1-e^{-\frac{x}{2l}})}\right] - 1}{1 + \sigma_0^{-1}\beta_1\gamma(\frac{l_y}{2} - |y|)\sin\alpha}.\tag{10}$$

In the "collapse zone" of a goaf, due to the high degree of rock fragmentation, the change in the rock's broken expansion coefficient in the vertical direction is not obvious. It can be considered that the rock's expansion coefficient in a "collapse zone" will remain unchanged in the vertical direction. Therefore, it is considered that the total void ratio

of a goaf, $\varphi_Z (x, y)$, is the difference between the broken expanded void ratio and the overburden void ratio, as shown in Formula (9).

$$\varphi_Z(x,y) = \frac{-\sum\limits_{i=1}^{n} (h_{i+1} - h_i)\varphi_{i,i+1} + h_d\varphi_G}{\sum\limits_{i=1}^{n} (h_{i+1} - h_i)_i + h_d}. \tag{11}$$

### 2.2. Verification Methods

(1)    Verification of the numerical simulation method

We used the UDEC numerical model to verify the feasibility of the above calculation method and the result. The advantage of numerical simulation verification is the use of discrete element methods to truly reflect the shape of a goaf under the same geological conditions. The design model was 300 m × 87 m (length × height). The advancing distance of the working face was 200 m, the horizontal displacement was constrained by the horizontal boundary around the model, the vertical displacement was constrained by the bottom, and the upper boundary was free. The Coulomb–Moore model was used. Table 1 shows the parameters of each layer, and Table 2 shows the joint parameters. The above parameters were selected based on the actual test results of the adjacent working faces of the mine. The immediate roof strata were constructed by the Voronoi unit, as it could simulate the irregular fracture of the immediate roof.

**Table 1.** Parameters of the strata.

| No. | Strata | Height | Bulk (pa) | Shear (pa) | Tensile (pa) | Cohesion (pa) | Friction | Density (kg · m$^{-3}$) |
|---|---|---|---|---|---|---|---|---|
| 1 | Mudstone | 20 | $4.0 \times 10^9$ | $2.0 \times 10^9$ | $1.25 \times 10^6$ | $2.5 \times 10^6$ | 22° | 1600 |
| 2 | Coal | 4 | $2.5 \times 10^9$ | $1.8 \times 10^9$ | $1.17 \times 10^6$ | $2.0 \times 10^6$ | 23° | 1400 |
| 3 | Sandy mudstone | 2 | $6.0 \times 10^9$ | $3.0 \times 10^9$ | $1.5 \times 10^6$ | $3.0 \times 10^6$ | 24° | 2000 |
| 4 | Fine sandstone | 3 | $8.0 \times 10^9$ | $4.0 \times 10^9$ | $2.0 \times 10^6$ | $3.5 \times 10^6$ | 26° | 2500 |
| 5 | Sandy mudstone | 3 | $6.0 \times 10^9$ | $3.0 \times 10^9$ | $1.5 \times 10^6$ | $3.0 \times 10^6$ | 24° | 2000 |
| 6 | Fine sandstone | 3 | $8.0 \times 10^9$ | $4.0 \times 10^9$ | $2.0 \times 10^6$ | $3.5 \times 10^6$ | 26° | 2500 |
| 7 | Sandy mudstone 1 | 4 | $6.0 \times 10^9$ | $3.0 \times 10^9$ | $1.5 \times 10^6$ | $3.0 \times 10^6$ | 24° | 2000 |
| 8 | Sandy mudstone 2 | 6 | $6.0 \times 10^9$ | $3.0 \times 10^9$ | $1.5 \times 10^6$ | $3.0 \times 10^6$ | 24° | 2000 |
| 9 | Fine sandstone | 10 | $8.0 \times 10^9$ | $4.0 \times 10^9$ | $2.0 \times 10^6$ | $3.5 \times 10^6$ | 26° | 2500 |
| 10 | Medium sandstone | 12 | $12.0 \times 10^9$ | $5.0 \times 10^9$ | $3.0 \times 10^6$ | $3.8 \times 10^6$ | 28° | 2800 |
| 11 | Coarse sandstone | 20 | $14.0 \times 10^9$ | $6.0 \times 10^9$ | $3.2 \times 10^6$ | $4.0 \times 10^6$ | 27° | 2600 |

**Table 2.** Joint parameters.

| No. | Strata | Normal Stiffness (Gpa) | Shear Stiffness (Gpa) | Cohesion (MPa) | Internal Friction Angle (°) | Tensile Strength (MPa) |
|---|---|---|---|---|---|---|
| 1 | Mudstone | 2.5 | 1.5 | 0.6 | 17 | 0.6 |
| 2 | Coal | 0.4 | 0.2 | 0.3 | 12 | 0.9 |
| 3 | Sandy mudstone | 3.0 | 4.0 | 1.0 | 18 | 3.6 |
| 4 | Fine sandstone | 6.0 | 5.0 | 2.0 | 20 | 5.8 |
| 5 | Sandy mudstone | 3.0 | 4.0 | 1.0 | 18 | 3.6 |
| 6 | Fine sandstone | 6.0 | 5.0 | 2.0 | 20 | 5.8 |
| 7 | Sandy mudstone 1 | 3.0 | 4.0 | 1.0 | 18 | 3.6 |
| 8 | Sandy mudstone 2 | 3.0 | 4.0 | 1.0 | 18 | 3.6 |
| 9 | Fine sandstone | 6.0 | 5.0 | 2.0 | 20 | 5.8 |
| 10 | Medium sandstone | 8.0 | 6.0 | 2.3 | 22 | 7.60 |
| 11 | Coarse sandstone | 8.2 | 6.4 | 2.6 | 17 | 6.60 |

After the initial balancing of the model, an excavation was carried out every 10 m. The overlying strata's top and bottom displacement were monitored during the excavation. The numerical model is shown in Figure 3. After the program was run, we used the plot model command in UDEC to display the void space and used ImageJ software to analyze the void ratio in the diagram. The analysis method is shown in Figure 4.

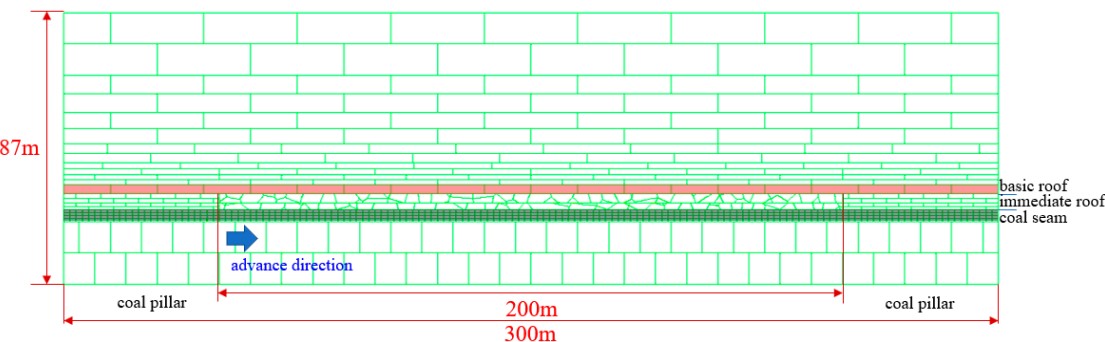

**Figure 3.** Numerical model.

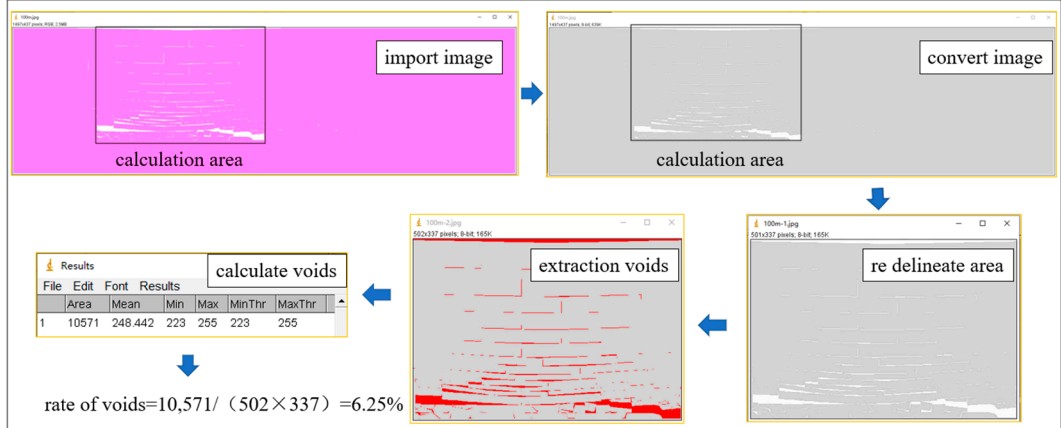

**Figure 4.** Graph processing method for the void ratio.

(2)  Verification of the experimental method

We further increased the similarity of the simulation validation to increase the reliability of the method. According to the similarity simulation rules, the geometric similarity ratio of the model was determined to be 80:1, the motion similarity ratio was 8.94:1, the stress similarity ratio was 100:1, and the mass similarity ratio was $0.64 \times 10^6$:1, with a time similarity ratio of 15:1. The design model was 2.5 m × 0.2 m × 0.8125 m (length × width × height), and the specific material ratio parameters for the similarity simulation are shown in Table 3.

**Table 3.** Proportions of the materials.

| No. | Strata | Ratio Number | Thickness (cm) | Dry Weight (kg) | Sand (kg) | CaCo$_3$ (kg) | Gypsum (kg) |
|---|---|---|---|---|---|---|---|
| 1 | Mudstone | 555 | 12.5 | 182.81 | 152.34 | 15.23 | 15.23 |
| 2 | Coal | 555 | 25 | 365.63 | 304.69 | 30.47 | 30.47 |
| 3 | Sandy mudstone | 773 | 5 | 73.13 | 63.98 | 6.40 | 2.74 |
| 4 | Fine sandstone | 337 | 2.5 | 36.56 | 27.42 | 2.74 | 6.40 |
| 5 | Sandy mudstone | 337 | 3.75 | 54.84 | 41.13 | 4.11 | 9.60 |
| 6 | Fine sandstone | 455 | 3.75 | 54.84 | 43.88 | 5.48 | 5.48 |
| 7 | Sandy mudstone 1 | 337 | 3.75 | 54.84 | 41.13 | 4.11 | 9.60 |
| 8 | Sandy mudstone 2 | 555 | 5 | 73.13 | 60.94 | 6.09 | 6.09 |
| 9 | Fine sandstone | 555 | 7.5 | 109.69 | 91.41 | 9.14 | 9.14 |
| 10 | Medium sandstone | 337 | 6.25 | 91.41 | 68.55 | 6.86 | 16.00 |
| 11 | Coarse sandstone | 337 | 6.25 | 91.41 | 68.55 | 6.86 | 16.00 |

We conducted design experiments to verify the distribution patterns of the voids and the formation process of the voids. A non-contact full-field strain measurement system, MatchID-2D, was used to monitor the voids. The system used a digital image correlation algorithm to provide displacement and strain data measurements in a two-dimensional field of view for the experiment. Using a digital image fixed-point capture analysis to

monitor the deformation of the physical model and taking photos every 10 min with a high-resolution camera at a fixed angle, we calculated and analyzed the displacement of each measuring point using image capture analysis software. The specific indicators that were monitored were ① the displacement changes on the surface, basic roof, and floor of the model during the mining process and ② the morphological changes in the development of the goaf. The physical similarity model is shown in Figure 5. The void fraction was calculated as the ratio of the difference in the vertical displacement between the surface and the floor to the mining height.

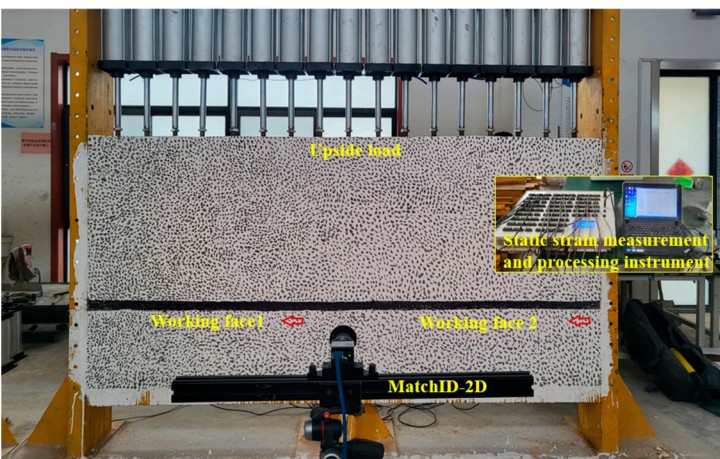

**Figure 5.** Physical similarity model.

### 3. Results

#### 3.1. Results of the Calculations

According to the actual geological occurrence conditions and the mining parameters of the coal seams mined by the working face of the mine, the average thickness of the coal seam was 4 m, the thickness of the direct roof was 5 m, and the average dip angle of the coal seam was 2°, which was regarded as nearly horizontal. The strike length of the working face was 200 m, the dip width was 100 m, and the mining height was 4 m. The roof was layered with fine sandstone and mudstone. The parameters of the rock strata are shown in Table 4. We substituted the parameters into Equations (3), (8) and (9) to obtain the void ratio distributions of the overburden deformation space, the collapse zone, and the total scope at any point, as shown in Figure 6.

It can be seen in Figure 6 that the void ratio in the middle of the stope reached 44.7% at the point close to the working face at the strike. As the rear roof was gradually compacted and stabilized, the void ratio decreased to 27.8%. The maximum void ratio at the stope boundary near the working face was 88.9%, and as the rear roof was gradually compacted and stabilized, the void ratio decreased to 33.3%. The turning point of the change in the void ratio outside the goaf was located 37 m behind the working face (the void ratio decreased by less than 1%), where the void ratio in the goaf decreased to below 45.4%. The turning point of the void ratio change in the middle of the goaf was 23 m behind the working face, where the void ratio in the goaf decreased to below 40.2%. The overall void ratio distribution of the goaf was from 27.8% to 33.3%.

In the inclined direction, the void fracture in the middle was low and high on both sides. The void fracture decreased from 33.9% to 29.3%, and the lowest point appeared in the middle. The void ratio of the boundary in the direction of the goaf incline showed a trend of high, middle, and low on both sides, from 88.9% to 44.8%, and the lowest point appeared in the middle of the goaf boundary. According to the void fracture distribution results and the geological mining conditions, the space in the working face that was available for filling was approximately 100,000~400,000 m$^3$ under this condition.

**Table 4.** Parameters of each overburden layer.

| No. | Buried Depth (m) | Stratum Thickness (m) | Distance from Stratum to Coal Seam (m) | Residual Broken Expansion Coefficient | Block Length of Broken Rock (m) |
|---|---|---|---|---|---|
| 1 | 700 | 5 | 0 | 1.005 | - |
| 2 | 708 | 3 | 8 | 1.007 | 5 |
| 3 | 711 | 3 | 11 | 1.012 | 8 |
| 4 | 715 | 4 | 15 | 1.024 | 11 |
| 5 | 721 | 6 | 21 | 1.049 | 12 |
| 6 | 731 | 10 | 31 | 1.078 | 15 |
| 7 | 743 | 12 | 43 | 1.089 | 18 |

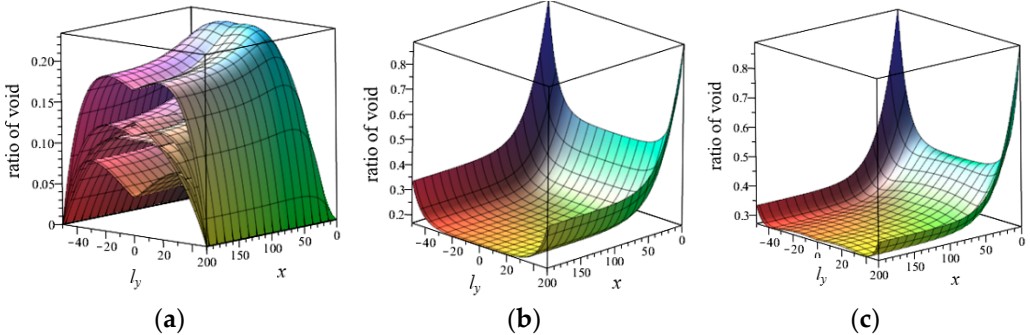

**Figure 6.** Void distribution in the goaf: (**a**) spatial distribution of the overlying rock deformation; (**b**) distribution of the void fraction in the collapse zone; and (**c**) distribution of the total void fraction.

*3.2. Results of the Verification*

First, we used a numerical model to analyze the accuracy of the calculation results. The overall void fraction of the model was 30.9%, which was within the range of 27.38–33.3% and was consistent with the theoretical calculation results. The distribution of the void fraction in the direction analysis model is shown in Figure 7a. This was consistent with the distribution trend of the void fraction along the goaf strike, as shown in Figure 6c.

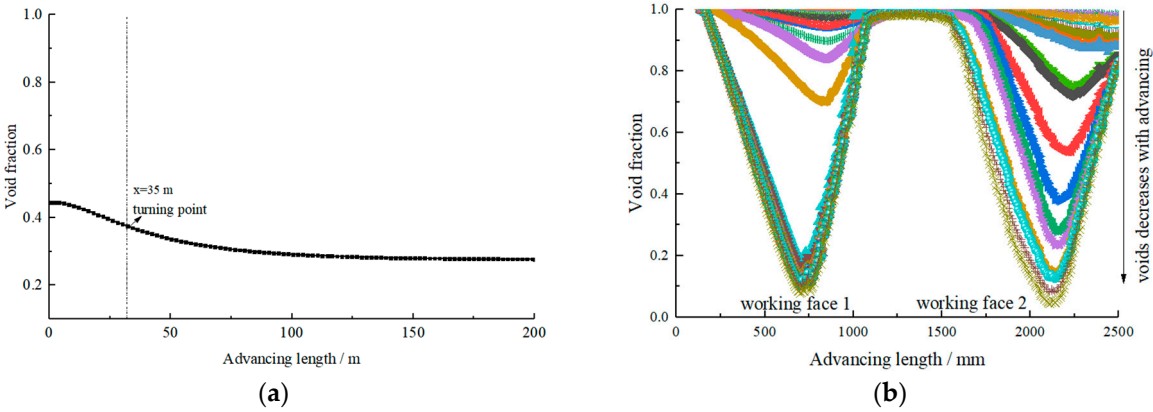

**Figure 7.** Distribution of the void fraction. (**a**) Verification of the numerical simulation method. (**b**) Verification of the experimental method.

In addition, the vertical displacement evolution curve of the surface of the similarity simulation model was obtained using matchID-2D, and the evolution of the void fraction in the goaf of the model was further obtained, as shown in Figure 7b. The total porosity of the model was calculated to be 28.0%, distributed between 27.38% and 33.3%, verifying the accuracy of the theoretical calculation results. It was known that as the working face advanced, the overall void fraction gradually decreased from 98.2% to 28.2%.

## 4. Discussion

### 4.1. Key Factors of the Overall Void Fraction

From Formulas (3) and (9), it could be seen that the key parameters of the calculation model were the mining height, face length, number of overlying rock layers, and bulking coefficient of the crushed direct roof. Under fixed geological conditions, the influencing factors of face length and mining height were analyzed. The analysis was intended to obtain the distribution of the void fracture in the stope's strike and dip directions. We adjusted the working face length and the mining height, i.e., the $l_y$ and $H$ values, and analyzed their impacts on the void ratio of the stope, as shown in Figures 8 and 9. We set $l_y$ as 60 m, 80 m, 100 m, 120 m, and 150 m and set $H$ as 3 m, 4 m, 5 m, and 6 m.

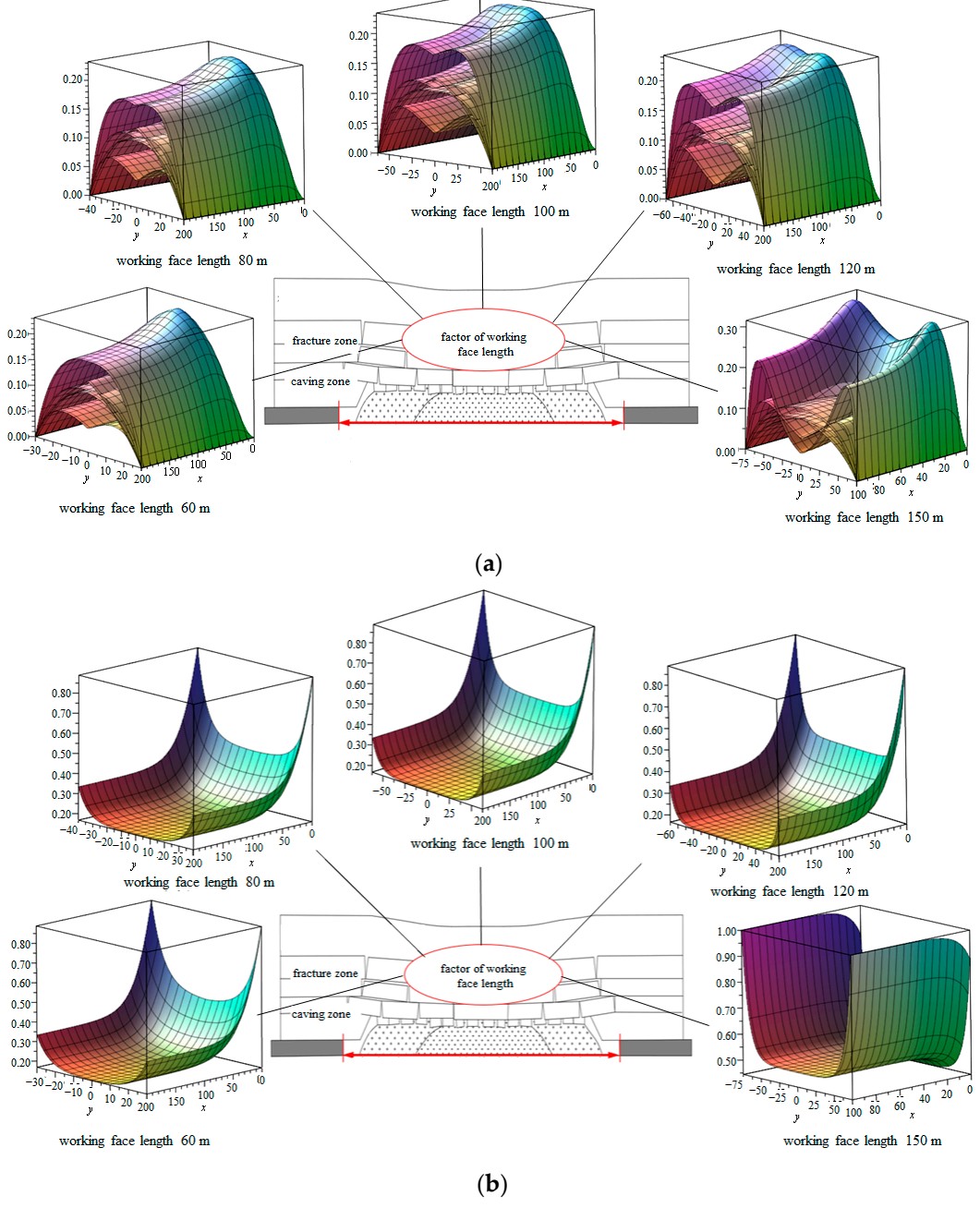

**Figure 8.** Void distribution of the stope under the influence of the face length. (**a**) Void distribution of the roof overburden. (**b**) Overall void distribution.

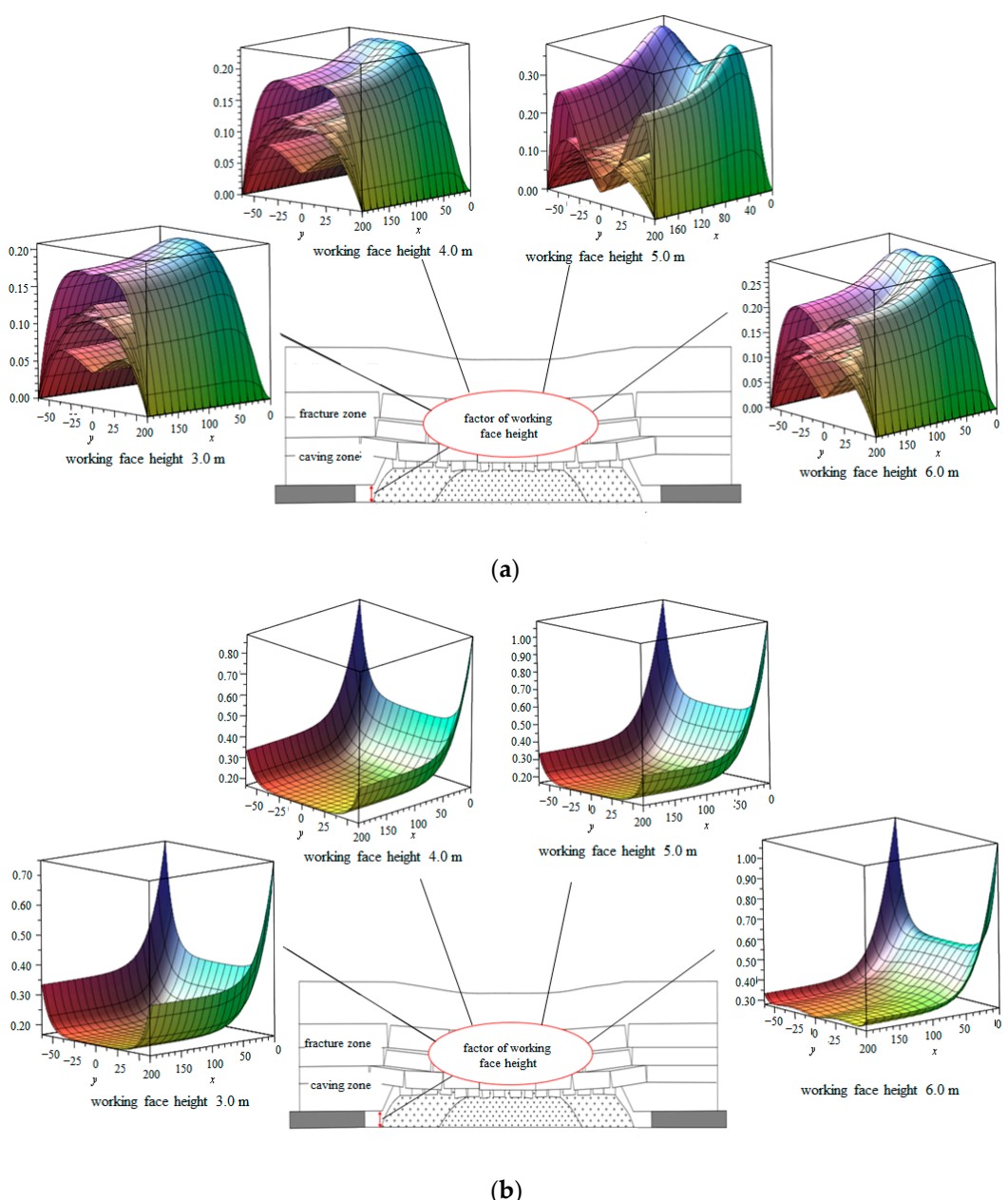

**Figure 9.** Distribution of the void ratio in the stope under the influence of the mining height. (**a**) Void distribution of the roof overburden. (**b**) Overall void distribution.

The face length impacted the void ratio in the middle strike of the goaf, and the maximum and minimum values of the void ratio in the middle gradually decreased with the increase in the face length from 46.7% and 42.3% to 44.4% and 40.9%, respectively. The turning point of the porosity in the middle remained unchanged at 20~23 m. The face length impacted the distribution of the void ratio in the goaf towards the middle. With the increase in the face length, the minimum value of the void ratio in the middle and the minimum value at the boundary gradually decreased from 30% and 46.7% to 29.2% and 44.5%, respectively.

According to the analysis in Figure 9, the mining height affected the strike porosity of the goaf. With the increase in mining height, the maximum porosities of the middle and the boundary increased from 37.5% and 75.0% to 54.6% and 100%, respectively, and the minimum porosities of the middle and the boundary were not affected. The mining height affected the porosity tendency of the goaf, and the mining height increased. The maximum porosities of the middle and the boundary increased from 33.7% and 75.0% to

34.3% and 100%, respectively. The minimum values of the void fractions in the middle and the boundary increased from 26.6% and 37.5% to 30.3% and 54.6%, respectively.

### 4.2. Evolution Law of the Void Fraction While Advancing

(1) Evolution of the void fraction

First, we analyzed the evolution law of the overall void when the working face advanced from 20 m to 200 m. The development of the two zones and the voids in the working face advancement were analyzed. The detailed changes in the void fraction during the advancement are shown in Figure 10. With the advancement of the working face, the void area gradually increased, and the void ratio gradually decreased from 70.1% to 25.1%. At the beginning of the mining of the working face, there was a large void ratio behind the working face that was close to 70%. From this analysis, it could be seen that as the working face advanced, the overall void ratio gradually decreased, and the overall void area first increased and then decreased.

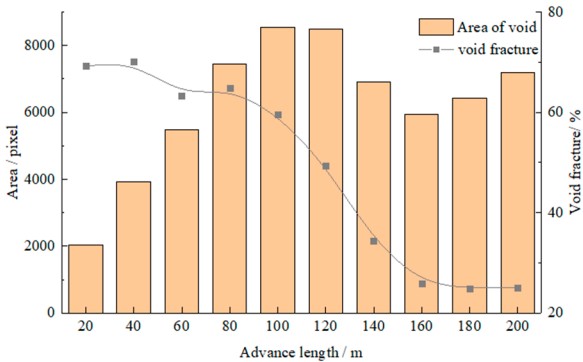

**Figure 10.** Change trend in the overall void ratio in the goaf during the process of advancing.

In addition, during the advancement of the working face, the rear void was constantly in a changing state. For example, when the working face advanced by 50 m, the distribution of the voids within 20 m behind the working face was completely different from when the working face had advanced by 60 m. The following is an analysis of the distributions at different depths inside the goaf during the advancement of the working face, as shown in Figure 11. According to the analysis in Figure 11a, before the working face was pushed to 100 m, the overall void was relatively large. At the same advancement, the voids in the goaf behind the working face exhibited a pattern of more voids on both sides and less voids in the middle, similar to a U-shaped distribution. At different advancements, the maximum value of the void appeared 20 m behind the working face, and as the distance of the advancement increased, the void fraction at that location gradually decreased. Figure 11b shows that after the working face was pushed to 100 m, the void behind the working face gradually decreased.

(2) Evolution of the void shape

We analyzed the evolution law of the void shape in the goaf using numerical and similarity models, as shown in Figure 12. Based on the analysis in Figures 7b and 12b, it could be seen that when the working face advanced to the initial breaking distance of the direct roof, a separation layer appeared, and at this time, the overall void fraction was extremely high, ranging from 95% to 100%. Afterwards, the direct roof periodically broke, and the void fraction decreased to below 88.9%. When the initial breaking distance of the basic roof was reached, it was obvious that the suspended roof and the masonry beam B block and masonry beam C block structures could be observed in the goaf. The basic roof, the suspended roof, and the internal rock blocks intersected and overlapped into a triangular area, forming a natural accumulation area, and the overall void fraction further decreased to below 33.3%. After reaching the basic roof and the periodically

breaking length, a natural accumulation area and a compaction area formed inside the goaf. Under the gravity of the overlying rock, the internal voids in the re-compaction area were relatively small, accounting for 33.0% of the total goaf, and the overall void fraction further decreased to below 27.8%. As the pushing distance further increased, the length of the natural accumulation area and the re-compaction area increased proportionally.

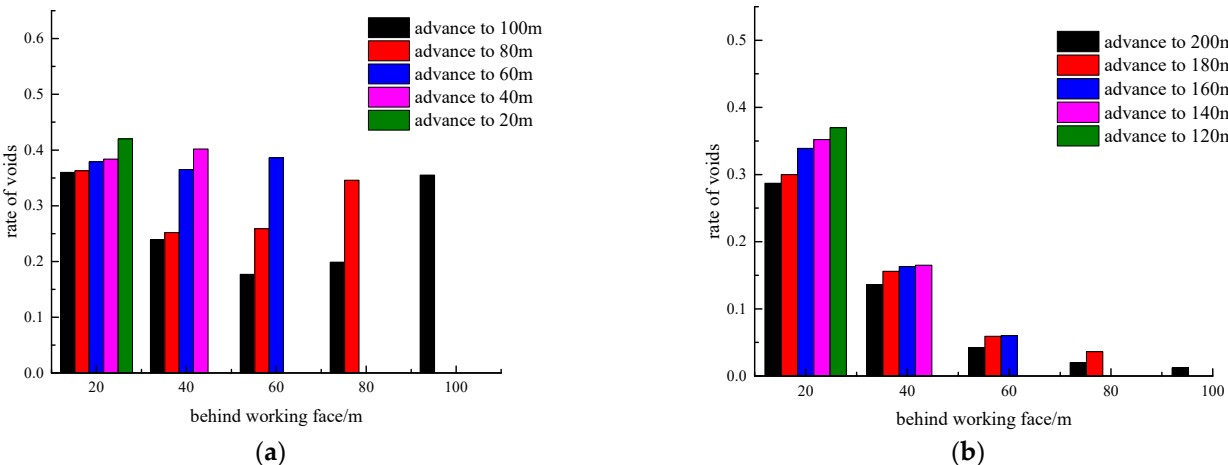

**Figure 11.** Void distribution behind the working face. (**a**) Before the working face advanced 100 m. (**b**) After the working face advanced 100 m.

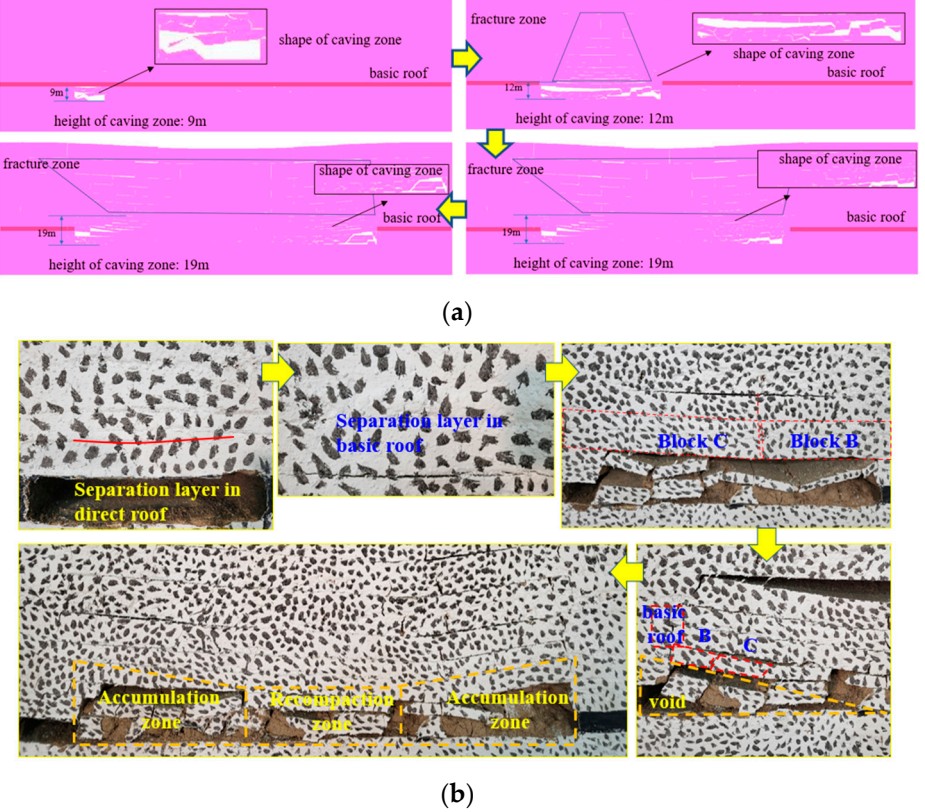

**Figure 12.** Evolution of the void's shape: (**a**) the numerical model and (**b**) the similarity model.

### 4.3. Divisions of the Types of Voids and Future Research

(1)     Divisions of the types of voids

Based on the explorations of the void fractions and void shapes, the overburden of the stope was vertically divided into a collapse zone, a fracture zone, and a bending subsidence zone, and the coal seam was horizontally divided into a natural accumulation zone and a re-compaction zone. The voids in the goaf were widely distributed in an accumulation zone. The mining influence line bounded the void's distribution area, namely, class I and class II, as shown in Figure 13.

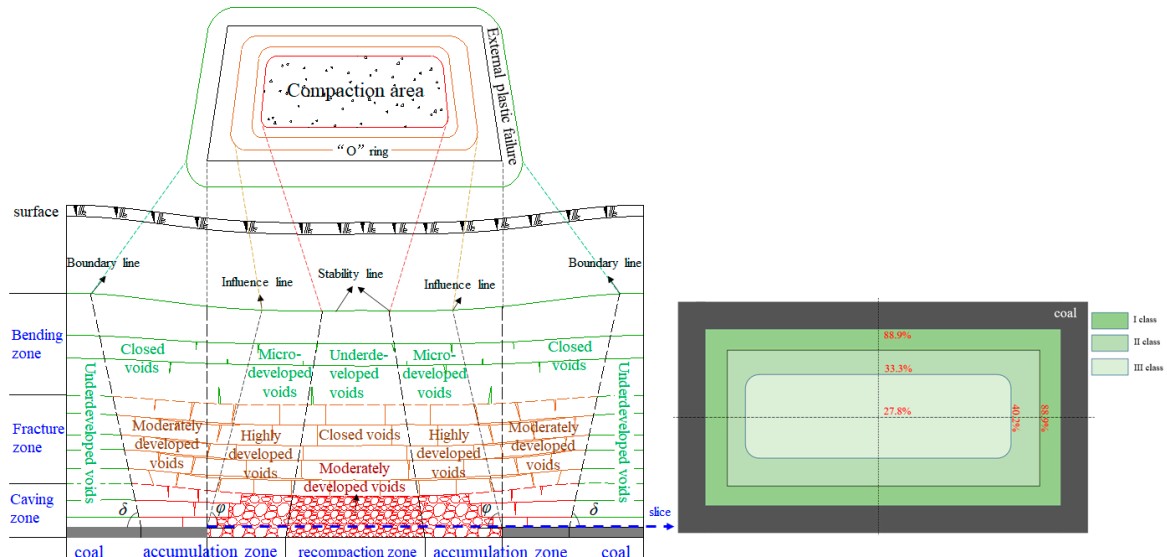

**Figure 13.** Void distribution in the goaf.

(2)     Future research

The above analysis shows that many voids existed behind the working face after mining. Voids are a resource that can be used to treat solid wastes such as gangue. In recent years, many scholars have proposed methods for gangue treatment to achieve green mining and improve coal quality [33–35]. The fluidization filling method for gangue has been put forward and was inspired by the above research. The technical concept of this method is to separate coal gangue underground, crush the washed and excavated gangue underground, and mix it with water to prepare fluidized gangue. The fluidized gangue is pumped to the working face through a filling pump and a pipeline, and it is inserted into the gap in the goaf using pipelines. Under the condition that the production efficiency of the working face is not affected, the underground gangue treatment capacity can be maximized. The overall concept of gangue fluid filling is shown in Figure 14. This technology can reduce the cost of gangue treatment without affecting normal production. At the same time, it can reduce the ecological damage and pollution of the ground environment and prevent spontaneous combustion near the goaf and the stop line.

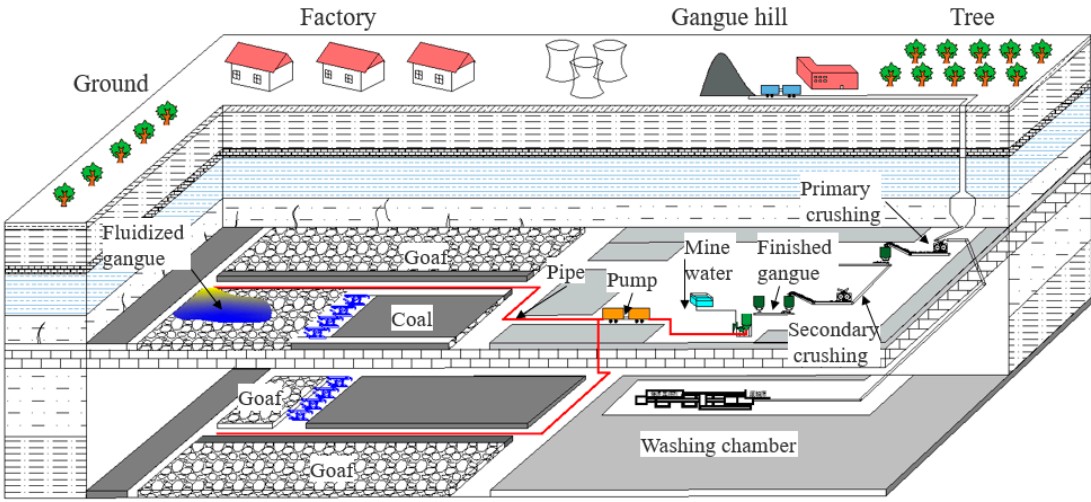

**Figure 14.** Fluidized filling method for waste rock.

## 5. Conclusions

This study explored the distribution law of a large number of voids generated during coal mining, summarized the range of the voids around and in the middle of the stope, determined the change nodes of the voids, and drew the following conclusions through theoretical calculations and simulation verification analyses:

(1) The calculation model of the stope void was established, and it was found that the void ratio at the stope boundary was far greater than that at the middle. The void ratio at the stope boundary in the strike direction gradually decreased, with the maximum value reaching 88.9% and the minimum value reaching 27.8%. The porosity at the stope boundary in the dip direction was U-shaped, with a maximum value of 88.9% and a minimum value of 33.3%.

(2) Numerical and similarity models were established to verify and describe the development shape of the gap between the two zones and the dynamic development of the void ratio during the advancement process. With the advancement of the working face, the void ratio in the stope gradually decreased from 70.1% to 25.1%.

(3) Before the working face advanced to 100 m, the void in the goaf behind the working face was evenly distributed, showing a pattern of having more gaps on both sides and fewer voids in the middle. After the working face was pushed to 100 m, the overall gap decreased. The maximum value of the void ratio occurred within 20 m behind the working face when it was pushed only 120 m, reaching 0.37.

(4) A distribution map of the void ratio in the stope was constructed. It was found that there were many voids in the collapse zone, with an average void ratio of nearly 40%. A 'gangue fluidization filling method' has been proposed to utilize these voids.

**Author Contributions:** Conceptualization and writing—original draft preparation, J.W.; Validation, N.Z. and G.M.; Investigation, supervision, and funding acquisition, M.L.; Data curation, C.W. All authors have read and agreed to the published version of the manuscript.

**Funding:** This work was supported by the National Natural Science Foundation of China (52004271 and 52130402), the China Postdoctoral Science Foundation (2021M693417), and the Jiangsu Postdoctoral Research Funding Program (2021K039A).

**Informed Consent Statement:** Not applicable.

**Data Availability Statement:** The data used to support the findings of this study are included within the article.

**Conflicts of Interest:** The authors declare no conflict of interest.

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
