# Peer review of "Distribution and Evolution Law of Void Fraction in the Goaf of Longwall Mining in a Coal Mine: Calculation Method and Numerical Simulation Verification"

_applsci, doi:10.3390/app13126908_

Round 1

Reviewer 1 Report

1. The reference list is absolutely unbalanced - only the papers from China are used

2. The void ratio definition is not clear and the size of the voids under study was not described

3. The experimental confirmation of the results was not given.

Must be improved

Author Response

Dear reviewer,

Reviewer 2 Report

Comment 1.

The paper examines the formation and behavior of voids in coal mining. A theoretical model is developed to understand how voids are distributed in the mined area, and this model is validated through numerical calculations. The study finds that voids are widely spread around the excavation site, and their distribution is influenced by factors like mining height and face length. As mining progresses, the voids initially increase and then decrease in size. The research also identifies specific void ratios in different geological conditions. Additionally, the paper highlights the potential use of these voids for backfilling mine waste and protecting the environment.

Overall, this study contributes to the knowledge and understanding of void development in mining processes, offering practical implications for optimizing mining conditions, utilizing voids for waste management, and protecting the environment. The findings and models developed in this research provide practical tools and insights that can be applied to optimize mining operations and enhance environmental protection.

Comment 2.

The paper could be challenging to comprehend, and it would be helpful if the authors had structured it using the conventional IMRaD format, which is widely known. By explicitly outlining the Methods, Results, and Discussion sections, the study could have followed a more traditional pattern that would make it easier for readers to grasp the main points of the manuscript.

Comment 3.

The Introduction section of the paper should be expanded to include a more extensive literature review. The current citation of only 20 sources is inadequate considering the breadth of this research. It would be advantageous for the authors to provide additional contextual information to elucidate the relevance of their study and acknowledge the limitations of their research. This would enable readers to gain a better understanding of the study's significance and the boundaries within which the conclusions should be interpreted.

Comment 4.

When expanding the Introduction section with an additional literature review, please take into account the suggested research from Ukraine and Vietnam. These studies may provide valuable insights that could enhance the overall quality of your paper.

Vu, T.T. Solutions to prevent face spall and roof falling in fully mechanized longwall at underground mines, Vietnam. Min. Miner. Depos. 2022, 16, 127-134. Doi:10.33271/mining16.01.127

Shavarskyi, Ia.; Falshtynskyi, V.; Dychkovskyi, R.; Akimov, O.; Sala, D.; Buketov, V. Management of the longwall face advance on the stress-strain state of rock mass. Mining of Mineral Deposits, 2022, 16, 78-85. https://doi.org/10.33271/mining16.03.078

Comment 5.

The calculation model and method must be described more preciously. Details are required. Describe key parameters.

Comment 6.

Equations 1-11. Please check if all symbols are given in text.

Comment 7.

Figure 3-5. Sharper quality required.

Comment 8.

Figure 9. Some legends are given not in English. It is difficult to understand for researchers out the China.

Comment 9.

Figure 10 must be discussed for better understanding the received research results.

Comment 10.

What factors were found to influence the overall void fraction in the goaf during the mining process?

Comment 11.

Were there any significant findings or patterns observed regarding the dynamic development and evolution of voids as the working face advanced in the longwall mining process?

Comment 12.

Figure 11. How Classes I-II were received? What were the key factors here?

Comment 13.

“Evolution Law” is given only in the Title of the paper. As it is important in this paper, the authors need to provide this information in the paper.

It would be beneficial if the paper explored the impact of leaving rock underground after coal mining on the Damage Evolution Law. It is worth considering the research mentioned below, as it could influence the direction of future research. This additional information could provide a more comprehensive understanding of the factors that affect Damage Evolution Law and may have implications for the design and operation of coal mines.

Smoliński, A.; Malashkevych, D.; Petlovanyi, M.; Rysbekov, K.; Lozynskyi, V.; Sai, K. Research into Impact of Leaving Waste Rocks in the Mined-Out Space on the Geomechanical State of the Rock Mass Surrounding the Longwall Face. Energies, 2022, 15, 9522. https://doi.org/10.3390/en15249522

Malashkevych, D.; Petlovanyi, M.; Sai, K.; Zubko, S. Research into the coal quality with a new selective mining technology of the waste rock accumulation in the mined-out area. Min. Miner. Depos. 2022, 16, 103-114. https://doi.org/10.33271/mining16.04.103

Comment 14.

Please provide a brief overview of the proposed future research.

Comment 15.

I am thoroughly impressed with the quality of this study and would highly recommend it for publication after some thoughtful revisions. The research conducted here is of great value and has the potential to make a substantial contribution to the field of mining. With the suggested improvements and clarifications, this research will undoubtedly capture the interest of a diverse readership.

Author Response

Dear reviewer,

Round 2

Reviewer 1 Report

The authors significantly improved the Ms and I recommend to accept it for publication

Reviewer 2 Report

Dear authors, you have provided a very good revision.
Without any doubt, I will recommend your paper for publication.